# Molecular Alterations in TP53, WNT, PI3K, TGF-Beta, and RTK/RAS Pathways in Gastric Cancer Among Ethnically Heterogeneous Cohorts

**DOI:** 10.3390/cancers17071075

**Published:** 2025-03-23

**Authors:** Cecilia Monge, Brigette Waldrup, Francisco G. Carranza, Enrique Velazquez-Villarreal

**Affiliations:** 1Center for Cancer Research, National Cancer Institute, Bethesda, MD 20892, USA; 2Department of Integrative Translational Sciences, Beckman Research Institute, City of Hope, Duarte, CA 91010, USA; 3City of Hope Comprehensive Cancer Center, Duarte, CA 91010, USA

**Keywords:** gastric cancer, cancer disparities, genetic mutations, precision medicine, TP53 pathway, WNT pathway, PI3K pathway, TGF-Beta pathway, RTK/RAS pathways

## Abstract

Gastric cancer (GC) is a major cause of cancer-related deaths, but not all patients are affected the same way. Hispanic/Latino (H/L) patients experience unique molecular changes compared to Non-Hispanic White (NHW) patients, yet research on these differences is limited. This study examines key cancer-related pathways to identify differences in mutation frequencies and their potential impact on survival. We found that H/L patients had fewer TP53 mutations but more WNT pathway alterations compared to NHW patients. While survival differences were significant in NHW patients with TP53 and PI3K pathway mutations, they were not observed in H/L patients. These findings highlight the importance of considering genetic and ethnic differences in GC research. Understanding these distinctions can help improve personalized treatment strategies and address disparities in cancer care.

## 1. Introduction

Gastric cancer (GC) remains one of the most prevalent and lethal malignancies worldwide, ranking as the fourth leading cause of cancer-related deaths globally [1,2]. Although advances in diagnostic tools and treatment strategies have improved survival rates in high-income countries, GC continues to disproportionately impact specific racial and ethnic populations [3]. Notably, Hispanic/Latino (H/L) individuals face a higher incidence of GC and worse survival outcomes compared to Non-Hispanic White (NHW) patients [4,5,6,7]. These disparities persist even after adjusting for socioeconomic factors, suggesting underlying biological differences that warrant further investigation [8,9,10]. While previous studies have explored the molecular landscape of GC in Asian and NHW populations, genomic alterations specific to the H/L population remain largely underexplored.

Molecular profiling studies have identified five key signaling pathways involved in GC development and progression: TP53, WNT, PI3K, TGF-Beta, and RTK/RAS. These pathways regulate essential cellular functions such as proliferation, apoptosis, DNA repair, and immune response, and their dysregulation is frequently associated with aggressive tumor behavior, therapy resistance, and poor prognosis [11,12,13]. Understanding pathway-specific alterations in H/L GC patients is crucial for uncovering ethnicity-specific molecular drivers that contribute to disease progression and disparities in clinical outcomes.

The TP53 signaling pathway plays a pivotal role in genomic stability, apoptosis, and cell cycle regulation [14]. Mutations in TP53 are among the most common alterations in GC, with studies reporting mutation rates as high as 30–50% [15]. Helicobacter pylori (H. pylori) infection, a well-established risk factor for GC, has been implicated in TP53 mutations through genomic instability and DNA damage [16]. Furthermore, TP53 alterations have been linked to chemoresistance, potentially limiting treatment efficacy in GC patients [17]. However, the prevalence and prognostic impact of TP53 mutations in H/L GC patients remain poorly characterized.

The WNT signaling pathway, particularly the WNT/β-catenin axis, is a key regulator of cell proliferation and differentiation. Aberrant activation of this pathway has been observed in intestinal-type GC, where mutations in genes such as RNF43 (3–44%) and LRP1B (31–67%) drive tumor progression [18]. Studies suggest that WNT ligand overexpression in GC promotes cancer stem cell renewal and enhances invasive properties [19]. Given the high rates of late-stage diagnosis among H/L GC patients, it is essential to determine whether WNT pathway alterations contribute to more aggressive disease phenotypes in this population.

The PI3K/AKT/mTOR signaling pathway is a major driver of tumor metabolism, growth, and survival. PI3K pathway dysregulation, often through PIK3CA mutations and PTEN deletions, is associated with increased tumor invasiveness, immune evasion, and resistance to therapy [20]. In GC, PI3K/AKT pathway hyperactivation has been linked to poor prognosis and reduced response to chemotherapy [21]. Moreover, H. pylori infection has been shown to activate PI3K/AKT signaling, further underscoring its role in GC pathogenesis [22]. While PI3K/AKT pathway alterations have been extensively studied in other populations, their impact on GC outcomes in H/L patients remains unclear.

The TGF-Beta signaling pathway plays a paradoxical role in GC, functioning as both a tumor suppressor in cancer and a pro-oncogenic driver in advanced disease [23]. In later stages, TGF-Beta promotes epithelial-to-mesenchymal transition (EMT), angiogenesis, and immune suppression, leading to increased tumor progression and metastasis [24]. Elevated TGF-Beta1 expression has been correlated with poor survival in GC patients [25]. Additionally, H. pylori infection has been shown to activate the TGF-Beta pathway, contributing to tumor progression [26]. However, it is unknown whether TGF-Beta pathway alterations contribute to disparities in GC outcomes among H/L patients.

The RTK/RAS signaling pathway regulates key cellular processes, including growth, migration, and differentiation. Mutations in KRAS, NRAS, and BRAF are frequently observed in GC, particularly in intestinal-type tumors and metastatic patients [27]. These mutations drive the constitutive activation of MAPK and PI3K/AKT pathways, promoting tumor progression and drug resistance [28]. KRAS mutations have also been associated with resistance to EGFR-targeted therapies, which may have implications for treatment response [29] in H/L GC patients. However, the role of RTK/RAS pathway alterations in shaping the molecular landscape of H/L GC remains largely unstudied.

Given the increasing burden of GC in H/L populations and the limited molecular characterization of this disease in this group [7,20,21,22,23,24,25,26,27,28,29,30,31,32,33,34], this study aims to comprehensively analyze pathway-specific alterations in TP53, WNT, PI3K, TGF-Beta, and RTK/RAS signaling in GC. We compare mutation frequencies between H/L and NHW patients and evaluate the prognostic impact of these alterations on overall survival. By integrating genomic and survival data, this study seeks to provide novel insights into ethnicity-specific molecular differences that may contribute to GC disparities. These findings may inform precision medicine strategies and guide targeted therapeutic interventions to improve GC outcomes in underrepresented populations.

## 2. Materials and Methods

For our analysis, we leveraged clinical and genomic data from 11 gastric cancer (GC) datasets available through the cBioPortal database. These datasets encompassed studies categorized under GC, along with data from the GENIE BPC GC v2.0-public dataset. After selecting the datasets, we implemented specific filtering criteria to refine the samples. Patients were included if they were identified as H/L. This process resulted in two datasets meeting all criteria comprising 83 H/L GC patients. For NHW patients, 717 GC patients were included using the same inclusion criteria but applied within this specific racial and ethnic group (Table 1 and Table 2). This study represents one of the largest comprehensive characterizations of TP53, WNT, PI3K, TGF-Beta, and RTK/RAS pathway alterations in an underserved population, providing critical insights into the molecular disparities in GC.

Ethnicity classification (H/L and NHW) was based on dataset-provided annotations. We further stratified these groups based on the presence or absence of TP53, WNT, PI3K, TGF-Beta, and RTK/RAS pathway alterations, enabling a detailed examination of the interactions between ethnicity and these molecular changes. Table 1 presents the number of patients included in the analysis of H/L and NHW patients, with a total of 83 H/L patients and 717 NHW patients. This analysis evaluates the prevalence of TP53, WNT, PI3K, TGF-Beta, and RTK/RAS pathway alterations by comparing H/L and NHW GC patients. By integrating these stratifications, our study provides one of the most comprehensive characterizations of TP53, WNT, PI3K, TGF-Beta, and RTK/RAS pathway disruptions in an underserved population, offering valuable insights into potential molecular disparities and their implications for precision medicine in GC.

We conducted statistical analysis using Chi-square tests to assess the independence of categorical variables and explore potential associations between pathway alterations and ethnicity. This approach allowed us to assess whether certain molecular disruptions were more prevalent based on ethnicity, providing insights into patient heterogeneity and potential implications for treatment responses.

To evaluate overall survival, we conducted a Kaplan–Meier survival analysis, examining the impact of alterations in the TP53, WNT, PI3K, TGF-Beta, and RTK/RAS pathways. Kaplan–Meier survival curves were generated to depict survival probabilities over time, categorizing patients based on the presence or absence of these pathway disruptions. To assess statistical significance between survival curves, we applied the log-rank test. Additionally, median survival times were estimated, accompanied by 95% confidence intervals to ensure the reliability of these measures. This thorough analytical framework offered valuable insights into how specific pathway alterations influence patient outcomes among GC patients in the H/L population.

## 3. Results

From the cBioPortal projects mentioned above that reported ethnicity, we identified and constructed our H/L cohort, which comprised 83 samples, while the NHW cohort included 717 samples (Table 1). In terms of gender distribution, the H/L cohort consisted of 45.8% males and 54.2% females, while the NHW cohort had a slightly higher proportion of males, with 54.0% males and 46.0% females. Regarding tumor type, all cases in both cohorts were primary tumors, with no metastatic cases reported in either group. The primary tumor site differed between the two groups. In the H/L cohort, gastric tumors were found in 10.8% of cases, whereas, in the NHW cohort, gastric tumors were significantly more frequent at 27.1%. Small bowel tumors appeared in 3.6% of H/L patients and 15.5% of NHW patients. Tumors in the bowel (1.2%), jejunum (1.2%), and soft tissue (3.6%) were exclusive to the H/L cohort, whereas they were absent in the NHW group. For ethnicity classification, all patients in the H/L cohort identified as Spanish/Hispanic, whereas all patients in the NHW cohort identified as Non-Spanish, Non-Hispanic. Overall, the data indicate notable differences in tumor site distribution and gender proportions between the two cohorts, emphasizing potential ethnic-specific disparities in GC.

The comparative analysis of genomic features between H/L and NHW patients reveals notable distinctions (Table 2). The median mutation count was lower in the H/L cohort (two mutations, IQR: 1–2) compared to the NHW cohort (two mutations, IQR: 1–4), with a *p*-value of 0.01266, indicating a statistically significant difference. Similarly, the median TMB was lower in H/L patients (0.865 mutations/Mb, IQR: 0.865–1.73) compared to NHW patients (1.6 mutations/Mb, IQR: 0.9–2.2), with a highly significant *p*-value of 0.00175. These findings suggest that H/L patients exhibit a lower overall mutational burden in comparison to NHW patients. Additionally, the median FGA, which represents the fraction of the genome affected by copy number alterations, was lower in H/L patients (0.101, IQR: 0.03–0.15) than in NHW patients (0.142, IQR: 0.06–0.26), with a *p*-value of 0.002095. This suggests that H/L GC patients may have fewer structural alterations and chromosomal instability compared to NHW patients. Overall, these results highlight significant genomic differences between H/L and NHW GC patients, with H/L patients exhibiting lower mutation counts, TMB, and FGA, which may have implications for tumor biology and therapeutic responses in these populations.

Furthermore, within the WNT pathway, APC mutations were found to be significantly more prevalent in H/L GC patients compared to NHW patients (3.6% vs. 0.8%, *p* = 0.05) (Table 2). While no overall statistical significance was observed in the PI3K, TGF-Beta, and RTK/RAS pathways, several genes exhibited borderline significance, suggesting potential ethnic-specific variations in GC molecular profiles. In the PI3K pathway, alterations in EGFR (*p* = 0.07), FGFR1 (*p* = 0.05), FGFR2 (*p* = 0.05), and PTPN11 (*p* = 0.05) were observed, while SMAD4 (*p* = 0.08) exhibited borderline significance in the TGF-Beta pathway. These findings underscore the importance of ethnicity-specific molecular characterization in GC and may provide insights into pathogenesis and potential therapeutic targets for underrepresented populations.

In our analysis of genetic alterations in GC among H/L and NHW individuals, we observed notable differences in the frequency of TP53 and WNT mutations, with TP53 alterations reaching statistical significance (Table 3). TP53 mutations were present in 9.6% of H/L patients, compared to 19.0% of NHW patients (*p* = 0.03489), indicating a significantly lower prevalence of TP53 alterations in the H/L cohort. Conversely, the absence of TP53 mutations was more frequent in H/L patients (90.4%) compared to NHW patients (81.0%), further highlighting this disparity. Although WNT pathway alterations were not statistically significant, they were observed at a higher frequency in H/L patients (8.4%) compared to NHW patients (4.0%, *p* = 0.08674), suggesting a possible trend that warrants further investigation. PI3K pathway alterations were found in 9.6% of H/L patients and 15.1% of NHW patients (*p* = 0.2477), indicating no significant difference between the groups. Similarly, TGF-Beta pathway alterations were present in 2.4% of H/L patients and 1.4% of NHW patients (*p* = 0.3586), with no substantial disparity observed. Interestingly, RTK/RAS pathway alterations were highly prevalent in both cohorts, with 89.2% of H/L patients and 88.6% of NHW patients exhibiting these mutations (*p* = 1), suggesting a shared molecular signature in GC development across both ethnic groups. These findings highlight potential differences in the genetic landscape of GC between H/L and NHW patients, particularly with lower TP53 mutation prevalence in H/L individuals, which may have implications for tumor biology, disease progression, and targeted treatment approaches. Further research is necessary to explore the functional impact of these mutations and their potential role in ethnic-specific GC disparities.

The Kaplan–Meier survival analysis for H/L GC patients with TP53 pathway alterations indicated no statistically significant difference in overall survival between those with and without the alteration (Figure 1). A total of 14 patients were in the altered group, while 69 patients were in the not-altered group, with identical survival trajectories observed in both groups (*p* = 1). The overlapping confidence intervals highlight high variability in the survival estimates at different time points, suggesting that TP53 pathway alterations may not be a strong prognostic factor in this cohort. However, the small sample size may limit the statistical power, and further studies with larger datasets are needed to better assess the potential impact of TP53 alterations on GC outcomes.

Similarly, the Kaplan–Meier survival analysis for WNT pathway alterations showed no statistically significant difference in survival outcomes (Figure 1). Among the 19 patients with WNT pathway alterations, the survival curve remained relatively stable, with a single drop in survival occurring at approximately 30 months, followed by a plateau. The wide confidence intervals surrounding the survival curve suggest a high degree of uncertainty, likely due to the limited sample size. These findings indicate that WNT pathway alterations may not have a strong prognostic impact, though further validation with larger datasets is necessary to confirm these observations.

For PI3K pathway alterations, the Kaplan–Meier survival analysis revealed no statistically significant survival differences (Figure 1). A total of 19 patients were in the PI3K pathway-altered group, but no separate comparison group was available. The survival curve exhibited a drop in survival at approximately 30 months, followed by a plateau, with wide confidence intervals suggesting considerable variability. These findings indicate that PI3K pathway alterations may not serve as a strong prognostic marker in this cohort, though further investigation is warranted to assess their potential role in GC progression.

Similarly, the TGF-Beta pathway alterations displayed a comparable trend to the PI3K pathway (Figure 1). Among the 19 patients with TGF-Beta alterations, the survival curve showed a decline at approximately 30 months, followed by a plateau, with high variability due to the small sample size. No statistically significant survival differences were observed, reinforcing the need for larger studies to validate the potential prognostic implications of TGF-Beta pathway alterations in GC.

Lastly, the Kaplan–Meier survival analysis for RTK/RAS pathway alterations (Figure 1) showed no significant difference in survival between patients with and without alterations (*p* = 0.48). Of the patients analyzed, 7 were in the altered group, while 12 were in the not-altered group. The survival trajectories between both groups were nearly identical, and the overlapping confidence intervals suggest high uncertainty in survival estimates. Given the small sample size, additional research with larger cohorts is necessary to further explore the prognostic relevance of RTK/RAS pathway alterations in GC.

Overall, these findings emphasize the uncertainty in survival differences across multiple pathway alterations (Figure 1), underscoring the need for larger and more diverse datasets to better understand the molecular drivers of GC in H/L populations and their potential implications for targeted therapies.

Moreover, we generated Kaplan–Meier overall survival curves for Non-Hispanic White (NHW) gastric cancer (GC) patients, stratified by the presence or absence of TP53, WNT, PI3K, TGF-Beta, and RTK/RAS pathway alterations. The Kaplan–Meier survival analysis for TP53 pathway alterations in NHW GC patients indicated no statistically significant difference in overall survival between those with and without the alteration (Appendix A). A total of 14 patients were in the altered group, while 69 patients were in the not-altered group, and the survival trajectories of both groups appeared nearly identical (*p* = 1). The overlapping confidence intervals and minimal divergence in survival curves suggest that TP53 pathway alterations may not be a strong prognostic factor in NHW GC patients. However, given the small sample size of the altered group, the statistical power may be limited, potentially obscuring subtle survival differences. Further research with larger patient cohorts is necessary to clarify the role of TP53 alterations in this population.

In contrast, WNT pathway alterations in NHW GC patients showed a slight trend toward reduced survival compared to those without alterations (Appendix A). Among the 46 patients in the altered group and the 460 patients in the not-altered group, the survival curve of WNT-altered patients (red curve) exhibited a modest decline in survival probability relative to the not-altered group (blue curve). However, this difference was not statistically significant (*p* = 0.6), and the wide confidence intervals suggest a high degree of uncertainty, likely due to the smaller sample size in the altered group. While these findings do not provide strong evidence of WNT alterations affecting survival outcomes, further stratification or larger datasets may be necessary to assess their clinical relevance more comprehensively.

The Kaplan–Meier survival analysis for PI3K pathway alterations in NHW GC patients revealed a statistically significant association with poorer overall survival (Appendix A). Among the 97 patients in the altered group and the 602 in the not-altered group, patients with PI3K pathway alterations (red curve) had markedly worse survival outcomes compared to those without alterations (blue curve) (*p* < 0.0001). The clear separation between the survival curves supports the notion that PI3K pathway alterations may serve as a negative prognostic factor in NHW GC. The wider confidence intervals in the altered group suggest some variability in survival estimates, but the statistical significance underscores the potential impact of PI3K dysregulation on disease progression. These findings warrant further investigation into the clinical implications of PI3K alterations, particularly in the context of targeted therapeutic approaches.

Similarly, TGF-Beta pathway alterations were associated with significantly worse survival outcomes in NHW GC patients (Appendix A). Among the 56 patients with TGF-Beta pathway alterations and the 486 in the not-altered group, patients harboring TGF-Beta pathway mutations (red curve) demonstrated a noticeable decline in survival probability compared to those without alterations (blue curve). This difference was statistically significant (*p* = 0.016), suggesting that TGF-Beta pathway dysregulation may contribute to disease progression and worse prognosis. However, the wide confidence intervals around the altered group’s survival curve indicate variability in survival estimates, likely due to the small sample size. Further research is needed to assess the biological mechanisms underlying these alterations and their potential as therapeutic targets in NHW GC.

In contrast, RTK/RAS pathway alterations did not show a statistically significant impact on overall survival in NHW GC patients (Appendix A). Among the 57 patients in the altered group and the 517 in the not-altered group, the survival trajectories of both groups were nearly identical, with no significant difference observed (*p* = 0.99). The overlapping survival curves and wide confidence intervals suggest that RTK/RAS pathway alterations may not play a major prognostic role in this cohort. However, given the small number of altered cases, additional studies with larger datasets may help determine whether RTK/RAS alterations influence survival outcomes in specific GC subtypes.

In summary, these results highlight pathway-specific differences in survival outcomes among NHW GC patients, with PI3K and TGF-Beta pathway alterations showing significant associations with poorer prognosis, whereas TP53, WNT, and RTK/RAS alterations did not appear to strongly influence survival outcomes. These findings underscore the importance of genomic profiling in NHW GC patients and suggest that targeting PI3K and TGF-Beta pathways may offer potential therapeutic benefits. Further studies with larger and more diverse patient populations are needed to better understand the clinical implications of these molecular alterations and their potential role in precision medicine strategies for GC treatment.

The alteration rates of TP53, WNT, PI3K, TGF-Beta, and RTK/RAS pathway-related genes were analyzed among H/L and NHW GC patients to determine potential ethnicity-related differences (Appendix A). The analysis revealed that most genes associated with the TP53 pathway, including MDM2, MDM4, RPS6KA3, and CHEK2, displayed low alteration rates in both H/L and NHW patients, with no statistically significant differences. TP53 mutations were slightly more prevalent in H/L patients (6.0%) compared to NHW patients (4.5%), but this difference was not statistically significant (*p* = 0.5765). Similarly, alterations in ATM and CDKN2A were observed at low frequencies in both groups, with no significant differences detected. These findings suggest that TP53 pathway mutations are relatively rare in both populations, and their role in GC progression may be influenced by additional molecular or environmental factors.

Within the WNT pathway, APC mutations were found to be more frequent in H/L GC patients (3.6%) compared to NHW patients (0.8%), showing a borderline statistical significance (*p* = 0.05708). Other WNT pathway-related genes, including CTNNB1, AXIN2, and TCF7L2, exhibited low alteration rates with no significant differences between the two groups. These findings suggest that while APC mutations may be more common in H/L patients, further investigation is needed to determine their potential impact on tumor biology and disease progression.

For the PI3K pathway, PIK3CA and PTEN mutations were observed at higher frequencies in H/L patients (3.6%) compared to NHW patients (1.8%), but these differences were not statistically significant (*p* = 0.2257). Other PI3K-related genes, including PIK3R1, PIK3R2, AKT1, and MTOR, had low or absent mutation rates in both groups, with no significant differences. These findings suggest that PI3K pathway alterations may occur at comparable rates across ethnic groups, with no clear evidence of ethnicity-specific molecular drivers in this pathway.

In the TGF-Beta pathway, SMAD4 mutations were more prevalent in H/L patients (2.4%) compared to NHW patients (0.4%), with borderline statistical significance (*p* = 0.08645). However, other genes in this pathway, including TGFBR1, TGFBR2, SMAD2, and SMAD3, exhibited low or absent mutation rates in both groups, with no significant differences. The trend toward a higher frequency of SMAD4 mutations in H/L patients suggests a possible role in GC development, but larger datasets are needed to confirm this observation and determine its clinical significance.

Lastly, analysis of the RTK/RAS pathway revealed that EGFR mutations were more frequent in H/L patients (3.6%) compared to NHW patients (1.0%), showing a borderline significance (*p* = 0.07558). Similarly, FGFR1 and FGFR2 mutations were more prevalent in H/L patients (3.6% and 2.4%, respectively) compared to NHW patients (0.8% and 0.3%, respectively), with both showing borderline significance (*p* = 0.05708 and *p* = 0.05557, respectively). However, KRAS, NRAS, BRAF, and MET alterations did not show significant differences between the two groups, suggesting that mutations in key oncogenic drivers are largely similar across ethnicities.

These findings highlight potential ethnicity-specific trends in GC molecular alterations, particularly in the APC, SMAD4, EGFR, and FGFR1/2 genes. However, the lack of statistical significance in most comparisons underscores the need for larger studies to validate these observations and explore their implications for precision medicine and targeted therapy in H/L GC patients.

## 4. Discussion

GC remains a leading cause of cancer-related mortality worldwide, with significant racial and ethnic disparities in both incidence and outcomes. While extensive molecular profiling has been conducted in Asian and NHW populations, data on the genomic landscape of GC in H/L patients remain limited. This study aimed to characterize the molecular alterations in the TP53, WNT, PI3K, TGF-Beta, and RTK/RAS pathways in GC and explore their potential prognostic implications by comparing H/L and NHW patients. Our findings revealed ethnicity-specific differences in key molecular alterations, particularly within the TP53 and WNT pathways, with potential clinical implications for risk stratification and precision medicine approaches.

Our analysis revealed a significantly lower prevalence of TP53 mutations in H/L patients (9.6%) compared to NHW patients (19%, *p* = 0.03). TP53 plays a crucial role in maintaining genomic stability and preventing tumor progression. The lower mutation frequency in H/L patients suggests potential differences in the mechanisms of tumor development between the ethnic groups. Since TP53 alterations are frequently associated with poor prognosis, genomic instability, and therapy resistance, their reduced occurrence in H/L GC patients raises questions about alternative pathways driving tumorigenesis in this population. It is also possible that epigenetic modifications, rather than direct TP53 mutations, contribute to GC progression in H/L individuals, warranting further investigation through multi-omics analyses.

The lower prevalence of TP53 mutations in H/L GC patients compared to NHW patients may reflect distinct tumorigenic mechanisms in different ethnic populations. Our findings suggest that WNT pathway dysregulation, including a higher frequency of WNT alterations and APC mutations in H/L patients, could represent an alternative oncogenic driver in this group. Additionally, environmental exposures, dietary patterns, and Helicobacter pylori infection rates may contribute to ethnic-specific differences in GC pathogenesis. Previous studies have also indicated that microsatellite instability (MSI) and epigenetic modifications may play a more prominent role in certain populations, potentially influencing the observed TP53 mutation rates. Further research incorporating MSI analysis, epigenetic profiling, and functional studies will be necessary to clarify the biological basis of these disparities and their implications for GC progression and treatment strategies.

The WNT pathway alterations were more common in H/L patients (8.4%) than in NHW patients (4%, *p* = 0.08), suggesting a possible trend toward increased WNT pathway dysregulation in the H/L population. Specifically, APC mutations were significantly more prevalent in H/L patients (3.6% vs. 0.8%, *p* = 0.05), which may have implications for disease progression and tumor behavior. The WNT/β-catenin signaling pathway plays a central role in GC pathogenesis by regulating cell proliferation, differentiation, and stemness. The increased prevalence of APC mutations in H/L patients suggests that WNT pathway activation may serve as a distinct molecular driver in this population, which could influence tumor aggressiveness and treatment response. Previous studies have shown that WNT pathway activation correlates with increased cancer stem cell properties and chemoresistance, highlighting the potential need for targeted therapies aimed at this pathway in H/L GC patients.

While this study provides novel insights into ethnic-specific genomic disparities in GC, experimental validation is needed to confirm the functional impact of the observed alterations. Given the higher frequency of WNT pathway mutations in H/L patients, future studies should focus on validating the role of WNT signaling in GC progression within this population. Functional assays, including WNT activation studies, β-catenin nuclear localization analysis, and APC gene expression profiling, could provide deeper mechanistic insights into how this pathway contributes to tumorigenesis in H/L patients. Additionally, further research incorporating patient-derived models and single-cell transcriptomics could help elucidate the interplay between genomic alterations and tumor microenvironment factors in GC disparities. By integrating bioinformatics-driven discoveries with experimental validation, future work can refine precision medicine strategies for underrepresented populations.

Our findings highlight significant ethnic-specific differences in gastric cancer, particularly in TP53 and WNT pathway alterations. Given that WNT signaling is known to influence immune evasion by modulating T-cell infiltration and the tumor microenvironment, future studies should investigate the relationship between WNT activation and immune response in H/L patients. Additionally, TGF-Beta pathway alterations, including SMAD4 mutations, may play a role in shaping the immune landscape of GC. Further integration of immune profiling data, such as tumor immune infiltration scores and cytokine expression analyses, could provide a deeper understanding of how oncogenic pathways contribute to immune regulation in ethnically diverse GC patients. These insights could ultimately inform immunotherapy strategies tailored to specific populations.

While our study focused on mutation-based alterations in FGFR1, FGFR2, and other RTK-related genes, receptor tyrosine kinase amplifications are known to play a key role in GC pathogenesis. Prior studies [35] have demonstrated frequent RTK amplifications, including in FGFR, ERBB2, and MET, which contribute to oncogenic signaling. However, the publicly available dataset used in our analysis does not provide comprehensive copy number variation data for these genes. Future research incorporating CNV analysis will be essential to assess the role of RTK amplifications in GC disparities, particularly in underrepresented populations such as H/L patients, and determine their potential therapeutic implications.

Beyond genetic predispositions, environmental and dietary factors likely contribute to the observed disparities in GC between H/L and NHW populations. Prior studies have reported a higher prevalence of H. pylori infection among H/L individuals, a key risk factor for gastric carcinogenesis. Additionally, dietary differences, including higher consumption of salt-preserved foods and refined carbohydrates in H/L populations, may contribute to increased GC risk. Future research should integrate epidemiological, dietary, and genomic data to better understand the multifactorial nature of GC disparities and how these factors interact with molecular alterations in tumor development.

Our findings highlight a significant difference in TP53 mutation prevalence between H/L and NHW GC patients, suggesting possible ethnicity-specific mechanisms in tumorigenesis. While our analysis focused on overall mutation frequency, future studies should investigate whether distinct TP53 mutation spectra exist between these populations. Evaluating specific TP53 mutation types, hotspot regions, and their functional consequences could provide deeper insights into how these alterations influence tumor biology, prognosis, and treatment response. Given the current limitations of publicly available datasets in providing detailed mutation spectrum data across ethnic groups, prospective sequencing efforts focusing on diverse populations will be essential to further elucidate these differences.

Although alterations in the PI3K, TGF-Beta, and RTK/RAS pathways did not reach statistical significance, several genes within these pathways exhibited borderline significance, hinting at possible ethnicity-related differences in mutation profiles. Notably, within the PI3K pathway, genes such as EGFR (*p* = 0.07), FGFR1 (*p* = 0.05), FGFR2 (*p* = 0.05), and PTPN11 (*p* = 0.05) exhibited marginally higher alteration rates in H/L patients compared to NHW patients. These findings suggest that PI3K pathway dysregulation may still play a role in H/L GC, particularly through receptor tyrosine kinase (RTK) activation, even if the overall pathway alteration rate was not significantly different. Additionally, SMAD4 alterations in the TGF-Beta pathway (*p* = 0.08) were slightly more frequent in H/L patients, pointing to a potential role for TGF-Beta signaling in GC pathogenesis in this population. Given the dual role of TGF-Beta as both a tumor suppressor and a pro-oncogenic factor, further studies are needed to explore whether SMAD4 alterations contribute to differential disease progression between ethnic groups.

Kaplan–Meier survival analysis revealed ethnicity-specific differences in the prognostic impact of these pathway alterations. Among NHW GC patients, PI3K and TGF-Beta pathway alterations were significantly associated with worse overall survival (*p* < 0.0001 and *p* = 0.016, respectively), whereas no significant survival differences were observed for TP53, WNT, or RTK/RAS alterations. In contrast, H/L GC patients showed no significant survival differences based on pathway alterations. These findings suggest that PI3K and TGF-Beta pathway alterations may have a greater prognostic impact in NHW GC patients than in H/L patients. The reasons for this discrepancy remain unclear but may involve differences in tumor microenvironment, immune response, or genetic ancestry. Additionally, the smaller sample size of H/L patients may have limited the statistical power, underscoring the need for larger, well-powered studies to further investigate these potential prognostic differences.

The significant association between PI3K pathway alterations and poorer survival in NHW patients supports previous studies demonstrating that PI3K/AKT/mTOR signaling is a critical driver of tumor progression and therapy resistance in GC. Given that PI3K pathway alterations have been linked to resistance to EGFR-targeted therapies, our findings suggest that NHW GC patients with PI3K alterations may benefit from alternative therapeutic strategies targeting this pathway, such as PI3K inhibitors or combination therapies. Conversely, the lack of a significant survival impact in H/L patients suggests that additional molecular mechanisms may contribute to GC progression in this group, reinforcing the need for ethnicity-specific research.

Similarly, TGF-Beta pathway alterations were significantly associated with worse survival in NHW patients but not in H/L patients. This aligns with previous studies indicating that high TGF-Beta1 expression is linked to lower survival rates in GC patients. The TGF-Beta signaling pathway is known to promote epithelial-to-mesenchymal transition (EMT), immune evasion, and metastasis in advanced GC. The observed survival disparity raises intriguing questions about whether TGF-Beta pathway activation exerts distinct biological effects in H/L and NHW patients, potentially due to differences in immune response or tumor microenvironment.

Despite the strengths of this study, including one of the largest ethnicity-specific analyses of pathway alterations in GC, several limitations must be acknowledged. The retrospective nature of publicly available datasets introduces potential biases, including differences in sample collection, sequencing depth, and patient demographics. Furthermore, the underrepresentation of H/L patients in genomic databases remains a major challenge in cancer research. Our findings highlight the urgent need for larger, prospective studies focusing on H/L populations to fully characterize GC molecular disparities. Additionally, functional validation studies are required to determine whether the observed pathway alterations translate into meaningful biological differences in tumor progression and therapy response.

One drawback of our study is the relatively small number of H/L GC patients compared to NHW patients. This limitation highlights the broader issue of insufficient genomic data availability for underrepresented populations, such as H/L patients, in publicly accessible databases. Nevertheless, the recent update to the cBioPortal database has expanded the available dataset, allowing our study to be among the first to leverage this increased GC cohort for analysis. Despite this challenge, our findings offer valuable insights into molecular disparities in GC, including a lower frequency of TP53 mutations and a higher occurrence of WNT pathway alterations in H/L patients. Furthermore, the observed borderline significant differences in PI3K and TGF-Beta pathway-related genes point to potential ethnicity-specific biological mechanisms that require further exploration. These results emphasize the urgent need for larger and more diverse genomic datasets to confirm and extend these findings, ultimately fostering more equitable cancer research and advancing precision medicine for diverse populations.

A constraint of this study is the limited availability of detailed clinical variables, including treatment history, disease stage at diagnosis, and comorbidities, all of which play a significant role in shaping patient outcomes and survival. Although these factors are critical for clinical prognosis, publicly accessible datasets often lack completeness or consistency in their clinical annotations. Nevertheless, our study makes use of one of the few available databases that integrate both genomic and clinical data, enabling an investigation of molecular disparities in GC across diverse ethnic populations. Moving forward, future research incorporating more comprehensive clinical data from prospective cohorts will be crucial in further clarifying the relationship between molecular alterations and clinical outcomes, ultimately advancing precision medicine efforts for underrepresented groups.

Ethnicity classification in this study was based on dataset-provided annotations, which are typically derived from self-reported demographic data. While self-identified ethnicity is widely used in genomic studies, it does not fully capture the complexity of genetic admixture, particularly in H/L populations. Future research should incorporate genetic ancestry analysis to refine the understanding of molecular disparities in GC and explore how admixture influences tumor biology and therapeutic responses. Integrating ancestry-informative markers (AIMs) could provide a more precise characterization of genetic backgrounds and help tailor precision medicine approaches for diverse populations.

Our study provides one of the first ethnicity-focused analyses of molecular alterations in TP53, WNT, PI3K, TGF-Beta, and RTK/RAS pathways in GC, leveraging a comprehensive, publicly available dataset. The significant differences observed in TP53 and WNT pathway alterations between H/L and NHW patients emphasize the potential role of ethnicity-specific tumor biology. While the lack of experimental validation remains a limitation, our analysis offers novel insights into the molecular disparities in GC, underscoring the importance of considering genetic heterogeneity in precision medicine approaches. Future studies incorporating functional validation and additional patient cohorts will be essential to further delineate the biological implications of these findings and their impact on therapeutic strategies.

## 5. Conclusions

In conclusion, this study provides important insights into the molecular landscape of GC in H/L patients, revealing significant differences in TP53 and WNT pathway alterations compared to NHW patients. While TP53 mutations were significantly less frequent in H/L patients, APC mutations were more common, suggesting potential ethnicity-specific drivers of tumor progression. Additionally, PI3K and TGF-Beta pathway alterations were significantly associated with poor survival in NHW patients, but not in H/L patients, highlighting possible differences in the prognostic impact of these pathways across ethnic groups. These findings emphasize the importance of ethnicity-specific genomic research to better understand the biological underpinnings of GC disparities and inform the development of precision medicine strategies tailored to underrepresented populations. Future studies integrating multi-omics analyses, immune profiling, and clinical outcomes data will be essential to uncover the full extent of molecular differences and improve therapeutic interventions for H/L GC patients.

## Figures and Tables

**Figure 1 cancers-17-01075-f001:**
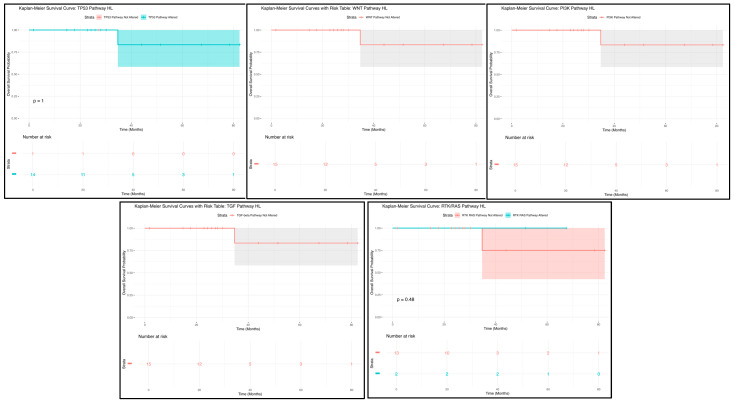
Kaplan–Meier overall survival curves for Hispanic/Latino (H/L) gastric cancer (GC) patients stratified by the presence or absence of TP53 (**upper left**), WNT (**upper middle**), PI3K (**upper right**), TGF-Beta (**lower left**), and RTK/RAS (**lower right**) pathway alterations.

**Table 1 cancers-17-01075-t001:** Patient demographics and clinical characteristics of the Hispanic/Latino (H/L) and Non-Hispanic White (NHW) gastric cancer (GC) cohorts.

Clinical Feature	H/L Cohort	NHW Cohort
	*n* (%)	*n* (%)
Gender		
Male	38 (45.8%)	387 (54.0%)
Female	45 (54.2%)	330 (46.0%)
Tumor Type		
Primary	83 (100.0%)	717 (100.0%)
Metastatic	0 (0.0%)	0 (0.0%)
Primary Tumor Site		
Gastric	9 (10.8%)	194 (27.1%)
Bowel	1 (1.2%)	0 (0.0%)
Small Bowel	3 (3.6%)	111 (15.5%)
Jejunum	1 (1.2%)	0 (0.0%)
Soft Tissue	3 (3.6%)	0 (0.0%)
NA	66 (79.5%)	412 (57.5%)
Ethnicity		
Spanish/Hispanic	83 (100.0%)	0 (0.0%)
Non-Spanish; Non-Hispanic	0 (0.0%)	717 (100.0%)

**Table 2 cancers-17-01075-t002:** Ethnicity-associated differences in clinical features between Hispanic/Latino (H/L) and Non-Hispanic White (NHW) gastric cancer (GC) cohorts.

Clinical Feature	H/L Samples n (%)	NHW Samples n (%)	*p*-Value
Median Mutation Count (IQR) *	2 (1–2)	2 (1–4)	0.01266
Median TMB (IQR) **	0.865 (0.865–1.73)	1.6 (0.9–2.2)	0.00175
Median FGA (IQR) ***	0.101 (0.03–0.15)	0.142 (0.06–0.26)	0.002095
Oncotree Code			
GIST	83 (100.0%)	717 (100.0%)	NA
APC Mutation			
Present	3 (3.6%)	6 (0.8%)	0.05708
Absent	80 (96.4%)	711 (99.2%)	
SMAD4 Mutation			
Present	2 (2.4%)	3 (0.4%)	0.08645
Absent	81 (97.6%)	714 (99.6%)	
EGFR Mutation			
Present	3 (3.6%)	7 (1.0%)	0.07558
Absent	80 (96.4%)	710 (99.0%)	
FGFR1 Mutation			
Present	3 (3.6%)	6 (0.8%)	0.05708
Absent	80 (96.4%)	711 (99.2%)	
FGFR2 Mutation			
Present	2 (2.4%)	2 (0.3%)	0.05557
Absent	81 (97.6%)	715 (99.7%)	
PTPN11 Mutation			
Present	2 (2.4%)	2 (0.3%)	0.05557
Absent	81 (97.6%)	715 (99.7%)	

* HL NA: 5, NHW NA: 48. ** HL NA: 66, NHW NA: 421. *** HL NA: 40, NHW NA: 391.

**Table 3 cancers-17-01075-t003:** Rates of TP53, WNT, PI3K, TGF-Beta, and RTK/RAS pathway alterations among Hispanic/Latino (H/L) and Non-Hispanic White (NHW) gastric cancer (GC) patients.

	H/L Samples n (%)	NHW Samples n (%)	*p*-Value
TP53 Alterations Present	8 (9.6%)	136 (19.0%)	0.03489
TP53 Alterations Absent	75 (90.4%)	581 (81.0%)	
WNT Alterations Present	7 (8.4%)	29 (4.0%)	0.08674
WNT Alterations Absent	76 (91.6%)	688 (96.0%)	
PI3K Alterations Present	8 (9.6%)	108 (15.1%)	0.2477
PI3K Alterations Absent	75 (90.4%)	609 (84.9%)	
TGF-beta Alterations Present	2 (2.4%)	10 (1.4%)	0.3586
TGF-beta Alterations Absent	81 (97.6%)	707 (98.6%)	
RTK/RAS Alterations Present	74 (89.2%)	635 (88.6%)	1
RTK/RAS Alterations Absent	9 (10.8%)	82 (11.4%)	

## Data Availability

All data used in the present study are publicly available at https://www.cbioportal.org/ (accessed on 31 January 2025) and https://genie.cbioportal.org (accessed on 31 January 2025). Additional data can be provided upon reasonable request to the authors.

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
