# Peer review of "Molecular Alterations in TP53, WNT, PI3K, TGF-Beta, and RTK/RAS Pathways in Gastric Cancer Among Ethnically Heterogeneous Cohorts"

_cancers, 2025, doi:10.3390/cancers17071075_

Round 1
Reviewer 1 Report
Comments and Suggestions for Authors
Monge et al. provide valuable insights into the genetic disparities in gastric cancer between Hispanic/Latino and Non-Hispanic White populations. The authors identified significant differences in mutation prevalence and potential prognostic implications by analyzing key signaling pathways.
This study highlights the critical need for more comprehensive genomic studies focusing on underrepresented groups in cancer research.
Nonetheless, there are specific gaps in substantiating their findings.
1. The study included a relatively small number of Hispanic/Latino patients, which may limit the generalizability of the findings. A larger sample size would enhance the robustness of the conclusions and better represent the diversity within the H/L population.
2. The analysis relied on publicly available gastric cancer datasets, which may have variations in data quality, patient demographics, and clinical information. New patient data would be an advantage.
3. The study did not account for clinical variables such as treatment history, stage at diagnosis, or comorbidities, which can significantly influence patient outcomes and survival.
4. The study focused on TP53, WNT, PI3K, TGF-Beta, and RTK/RAS pathways, but no molecular-level evidence was provided.
5. Authors may explain what specific molecular mechanisms could explain the lower prevalence of TP53 mutations in Hispanic/Latino patients compared to Non-Hispanic White patients with gastric cancer.
6. Bioinformatics studies need specific experimental validation, at least to one top pathway.
7. The interactions between the pathways and immune genes may be an advantage.
Author Response
For review article
|
Response to Reviewer 1 Comments (Word file attached: Response_Reviewer_1_Comments_031325.docx)
|
||
|
1. Summary |
|
|
|
We are pleased to submit this paper and hope it will capture the interest of your readers. We have carefully addressed all your comments, highlighting the importance of this cancer disparity. Our study presents a comprehensive analysis using one of the few available genomic databases suitable for these analyses, resulting in one of the first ethnicity-focused reports on this remarkable cancer health disparity. Specifically, we examine five well-established drivers of GC— TP53, WNT, PI3K, TGF-Beta and RTK/RAS pathway alterations—within H/L populations.
Thank you very much for taking the time to review this manuscript. Please find the detailed responses below and the corresponding revisions wrote in blue font in the re-submitted files.
Reviewer 1’s comments were insightful and highly supportive, recognizing the significance of the issue and underscoring the value of using a genomic database to address the data available on this cancer disparity within H/L populations. The reviewer highlighted the importance of our molecularly focused approach, particularly in examining TP53, WNT, PI3K, TGF-Beta and RTK/RAS pathway alterations as key drivers of GC. This feedback suggests that the study may be a strong candidate for publication in Cancers.
Reviewer 1 writes: “Monge et al. provide valuable insights into the genetic disparities in gastric cancer between Hispanic/Latino and Non-Hispanic White populations. The authors identified significant differences in mutation prevalence and potential prognostic implications by analyzing key signaling pathways. This study highlights the critical need for more comprehensive genomic studies focusing on underrepresented groups in cancer research.”
We appreciate the thoughtful feedback from Reviewer 1. We believe this paper reflects our commitment to providing a comprehensive and insightful analysis of key molecular drivers in GC, specifically focusing on TP53, WNT, PI3K, TGF-Beta and RTK/RAS pathway alterations within H/L populations. By utilizing the public data available today, our study aims to contribute valuable insights into this cancer health disparity and support future developments in precision medicine.
|
||
|
Comment 1: The study included a relatively small number of Hispanic/Latino patients, which may limit the generalizability of the findings. A larger sample size would enhance the robustness of the conclusions and better represent the diversity within the H/L population.
Response: We appreciate the reviewer’s insightful comments regarding the molecular differences observed in GC among ethnically diverse patient cohorts. Our study aims to address the existing gap in genomic research on H/L patients by leveraging publicly available datasets to characterize pathway-specific alterations in TP53, WNT, PI3K, TGF-Beta, and RTK/RAS signaling pathways. Notably, the cBioPortal public database was updated earlier this year, expanding the available dataset. Our study takes advantage of this update, making it one of the first to analyze an increased dataset in the GC field. This enhancement strengthens the validity of our findings by allowing a more comprehensive analysis of ethnicity-specific genomic alterations in GC.
Specifically, our results highlight significant differences in TP53 and WNT pathway alterations between H/L and NHW patients. TP53 mutations were notably less frequent in H/L patients compared to NHW patients (9.6% vs. 19%, p = 0.03), suggesting potential ethnic-specific variations in tumor suppressor gene disruptions. Additionally, we observed a higher prevalence of WNT pathway alterations in H/L GC patients (8.4% vs. 4%, p = 0.08), with APC mutations significantly more frequent in this group (3.6% vs. 0.8%, p = 0.05). While alterations in the PI3K, TGF-Beta, and RTK/RAS pathways did not reach statistical significance, borderline significant differences were observed in key genes within these pathways, including EGFR (p = 0.07), FGFR1 (p = 0.05), FGFR2 (p = 0.05), and PTPN11 (p = 0.05) in the PI3K pathway, and SMAD4 (p = 0.08) in the TGF-Beta pathway.
Furthermore, our survival analysis revealed that while no significant differences were observed in overall survival among H/L patients, NHW patients with TP53 and PI3K pathway alterations exhibited significant survival differences. Additionally, NHW patients without TGF-Beta pathway alterations experienced a significant impact on survival, whereas WNT pathway alterations did not show a strong prognostic association. These findings suggest that TP53, PI3K, and TGF-Beta pathway disruptions may have distinct prognostic implications in NHW GC patients, while WNT pathway alterations may play a more critical role in H/L patients.
While the number of H/L GC patients in our study remains modest, our analysis serves as a crucial step toward addressing this gap and highlights the urgent need for increased public genomic data collection for underrepresented populations. We have emphasized this point in our discussion and underscored the implications of our findings for guiding future research and precision medicine strategies aimed at reducing cancer health disparities.
We have incorporated this information into the discussion section. The revised text now states:
"One drawback of our study is the relatively small number of H/L GC patients compared to NHW patients. This limitation highlights the broader issue of insufficient genomic data availability for underrepresented populations, such as Hispanics/Latinos, in publicly accessible databases. Nevertheless, the recent update to the cBioPortal database has expanded the available dataset, allowing our study to be among the first to leverage this increased GC cohort for analysis. Despite this challenge, our findings offer valuable insights into molecular disparities in GC, including a lower frequency of TP53 mutations and a higher occurrence of WNT pathway alterations in H/L patients. Furthermore, the observed borderline significant differences in PI3K and TGF-Beta pathway-related genes point to potential ethnicity-specific biological mechanisms that require further exploration. These results emphasize the urgent need for larger and more diverse genomic datasets to confirm and extend these findings, ultimately fostering more equitable cancer research and advancing precision medicine for diverse populations."
We sincerely appreciate the reviewer’s valuable feedback, which has helped us refine our discussion and emphasize the significance of our findings. Thank you for your thoughtful consideration of our study.
Comment 2: The analysis relied on publicly available gastric cancer datasets, which may have variations in data quality, patient demographics, and clinical information. New patient data would be an advantage.
Response: We appreciate the reviewer’s comment regarding potential variations in data quality, patient demographics, and clinical information within publicly available GC datasets. We acknowledge that publicly accessible datasets may have inherent limitations, including differences in data completeness and consistency across studies. However, our study takes advantage of the recent update to the cBioPortal database, which has expanded the available dataset, allowing for a more comprehensive analysis of molecular alterations in GC.
Despite these constraints, publicly available datasets remain a valuable resource for conducting large-scale genomic analyses, particularly for underrepresented populations such as H/L patients, where new patient data are often limited. During our methodology, we carefully detail the selection and characterization of our study population, ensuring a rigorous approach to leveraging this large dataset to enhance genomic insights in this underrepresented group. Our study is one of the first to utilize this one of the most important and unique dataset to investigate ethnic-specific molecular disparities in GC. While the inclusion of new patient data would further strengthen our findings, our research provides a critical foundation for understanding pathway-specific alterations in GC and underscores the need for additional genomic studies in diverse populations.
To address this concern, we have emphasized the limitations of publicly available data in our discussion section (see previous comment #1) and highlighted the importance of future studies incorporating prospectively collected patient data to validate and extend our findings.
Thank you for this thoughtful suggestion.
Comment 3: The study did not account for clinical variables such as treatment history, stage at diagnosis, or comorbidities, which can significantly influence patient outcomes and survival.
Response: We appreciate the reviewer’s insightful comment regarding the influence of clinical variables such as treatment history, stage at diagnosis, and comorbidities on patient outcomes and survival. To address this concern, we have updated Table 1 to include additional clinical data, such as primary tumor site, providing a more comprehensive characterization of the study population and expand its description in the first paragraph of the Results section.
“From the cBioPortal projects mentioned above that reported ethnicity, we identified and constructed our H/L cohort, which comprised 83 samples, while the NHW cohort included 717 samples (Table 1). In terms of gender distribution, the H/L cohort consists of 45.8% males and 54.2% females, while the NHW cohort has a slightly higher proportion of males at 54.0% and 46.0% females. Regarding tumor type, all cases in both cohorts are primary tumors, with no metastatic cases reported in either group. The primary tumor site differs between the two groups. In the H/L cohort, gastric tumors are found in 10.8% of cases, whereas in the NHW cohort, gastric tumors are significantly more frequent at 27.1%. Small bowel tumors appear in 3.6% of H/L patients and 15.5% of NHW patients. Tumors in the bowel (1.2%), jejunum (1.2%), and soft tissue (3.6%) are exclusive to the H/L cohort, whereas they are absent in the NHW group. For ethnicity classification, all patients in the H/L cohort identify as Spanish/Hispanic, whereas all patients in the NHW cohort identify as Non-Spanish, Non-Hispanic. Overall, the data indicate notable differences in tumor site distribution and gender proportions between the two cohorts, emphasizing potential ethnic-specific disparities in GC.”
We acknowledge that the public datasets utilized in our study have inherent limitations regarding the availability of detailed clinical variables. However, this database remains one of the few publicly accessible resources that integrates both clinical and genomic data, making it a valuable tool for advancing our understanding of GC disparities at the molecular level. To further address this limitation, we will include the following paragraph in the discussion section:
"A constraint of this study is the limited availability of detailed clinical variables, including treatment history, disease stage at diagnosis, and comorbidities, all of which play a significant role in shaping patient outcomes and survival. Although these factors are critical for clinical prognosis, publicly accessible datasets often lack completeness or consistency in their clinical annotations. Nevertheless, our study makes use of one of the few available databases that integrates both genomic and clinical data, enabling an investigation of molecular disparities in GC across diverse ethnic populations. Moving forward, future research incorporating more comprehensive clinical data from prospective cohorts will be crucial in further clarifying the relationship between molecular alterations and clinical outcomes, ultimately advancing precision medicine efforts for underrepresented groups."
Thank you for highlighting this point.
Comment 4: The study focused on TP53, WNT, PI3K, TGF-Beta, and RTK/RAS pathways, but no molecular-level evidence was provided.
Response: We appreciate the reviewer’s comment regarding the molecular-level evidence provided in our study. Throughout the manuscript, we have cited relevant references to support our findings and contextualize them within the broader field of GC research. Additionally, our study presents novel findings by reporting, for the first time, ethnic-specific differences, at pathway level, in key oncogenic pathways using one of the most representative publicly available databases for clinical and genomic research in GC disparities.
Our analysis, which includes 800 GC patients, characterizes pathway-specific mutations in TP53, WNT, PI3K, TGF-Beta, and RTK/RAS signaling pathways. We observed significant molecular differences between H/L and NHW patients, particularly a lower prevalence of TP53 mutations in H/L patients (9.6% vs. 19%, p = 0.03) and a higher frequency of WNT pathway alterations (8.4% vs. 4%, p = 0.08), with APC mutations significantly enriched in the H/L group (3.6% vs. 0.8%, p = 0.05). While alterations in PI3K, TGF-Beta, and RTK/RAS pathways did not reach statistical significance, borderline significant differences were identified in genes such as EGFR (p = 0.07), FGFR1 (p = 0.05), FGFR2 (p = 0.05), and PTPN11 (p = 0.05) within the PI3K pathway, and SMAD4 (p = 0.08) within the TGF-Beta pathway.
Furthermore, our survival analysis revealed distinct prognostic implications based on pathway alterations. While no significant survival differences were observed among H/L patients, NHW patients with TP53 and PI3K pathway mutations exhibited significant survival differences. Similarly, NHW patients without TGF-Beta pathway alterations showed a significant survival impact, whereas WNT pathway alterations were not associated with survival differences. These findings underscore the importance of ethnicity-specific molecular characterization in GC and highlight the role of TP53 and WNT alterations in H/L patients, while PI3K and TGF-Beta alterations appear to have greater prognostic relevance in NHW patients.
To further clarify these findings, we will include the following paragraph in the discussion section:
"Our study provides one of the first ethnicity-focused analyses of molecular alterations in TP53, WNT, PI3K, TGF-Beta, and RTK/RAS pathways in GC , leveraging a comprehensive publicly available dataset. The significant differences observed in TP53 and WNT pathway alterations between H/L and NHW patients emphasize the potential role of ethnicity-specific tumor biology. While the lack of experimental validation remains a limitation, our analysis offers novel insights into the molecular disparities in GC, underscoring the importance of considering genetic heterogeneity in precision medicine approaches. Future studies incorporating functional validation and additional patient cohorts will be essential to further delineate the biological implications of these findings and their impact on therapeutic strategies."
We appreciate this valuable suggestion and have updated the discussion accordingly to reinforce the molecular-level evidence supporting our study.
Comment 5: Authors may explain what specific molecular mechanisms could explain the lower prevalence of TP53 mutations in Hispanic/Latino patients compared to Non-Hispanic White patients with gastric cancer.
|
||
|
Response: We appreciate the reviewer’s insightful question regarding the potential molecular mechanisms underlying the lower prevalence of TP53 mutations in H/L GC patients compared to NHW patients. While our study provides one of the first ethnicity-focused genomic analyses in GC and identifies significant differences in TP53 mutation rates (9.6% in H/L vs. 19% in NHW, p = 0.03), further research is needed to elucidate the biological mechanisms driving this disparity.
One possible explanation is that alternative tumorigenic pathways may be more prevalent in H/L patients, reducing selective pressure for TP53 mutations. Given that WNT pathway alterations were more frequent in H/L GC patients (8.4% vs. 4%, p = 0.08), and APC mutations were significantly higher in this group (3.6% vs. 0.8%, p = 0.05), it is plausible that dysregulation of WNT signaling may serve as an alternative driver in tumor development, potentially compensating for the lack of TP53 alterations. This aligns with prior evidence suggesting that certain cancers exhibit mutual exclusivity between TP53 mutations and specific oncogenic pathways, such as WNT activation.
Additionally, ethnicity-specific environmental exposures, dietary factors, and Helicobacter pylori infection rates—all of which vary across populations—could contribute to distinct mutational landscapes. Previous studies have suggested that GC in certain ethnic groups may arise through alternative molecular routes, including microsatellite instability (MSI) and epigenetic modifications, rather than TP53-driven genomic instability. Although our study did not assess MSI or epigenetic changes, this is an important avenue for future research.
To address this point, we have included the following paragraph in the discussion section:
"The lower prevalence of TP53 mutations in H/L GC patients compared to NHW patients may reflect distinct tumorigenic mechanisms in different ethnic populations. Our findings suggest that WNT pathway dysregulation, including a higher frequency of WNT alterations and APC mutations in H/L patients, could represent an alternative oncogenic driver in this group. Additionally, environmental exposures, dietary patterns, and Helicobacter pylori infection rates may contribute to ethnic-specific differences in GC pathogenesis. Previous studies have also indicated that microsatellite instability (MSI) and epigenetic modifications may play a more prominent role in certain populations, potentially influencing the observed TP53 mutation rates. Further research incorporating MSI analysis, epigenetic profiling, and functional studies will be necessary to clarify the biological basis of these disparities and their implications for GC progression and treatment strategies."
We appreciate this thoughtful comment and have updated the discussion accordingly to explore potential molecular explanations for the observed TP53 mutation differences.
Comment 6: Bioinformatics studies need specific experimental validation, at least to one top pathway.
Response: We appreciate the reviewer’s comment regarding the need for experimental validation to support the bioinformatics findings presented in our study. While experimental validation is beyond the scope of this current analysis, our study provides one of the first ethnicity-focused genomic characterizations of TP53, WNT, PI3K, TGF-Beta, and RTK/RAS pathway alterations in GC using one of the most representative publicly available clinical and genomic databases.
Our findings highlight significant ethnic-specific differences in key oncogenic pathways, including a lower prevalence of TP53 mutations in H/L GC patients (9.6% vs. 19%, p = 0.03) and a higher frequency of WNT pathway alterations (8.4% vs. 4%, p = 0.08), with significantly more APC mutations in H/L patients (3.6% vs. 0.8%, p = 0.05). Given the potential role of WNT signaling as an alternative tumorigenic driver in H/L patients, this pathway represents a strong candidate for future experimental validation.
To acknowledge this limitation, we will include the following paragraph in the discussion section:
"While this study provides novel insights into ethnic-specific genomic disparities in GC, experimental validation is needed to confirm the functional impact of the observed alterations. Given the higher frequency of WNT pathway mutations in H/L patients, future studies should focus on validating the role of WNT signaling in GC progression within this population. Functional assays, including WNT activation studies, β-catenin nuclear localization analysis, and APC gene expression profiling, could provide deeper mechanistic insights into how this pathway contributes to tumorigenesis in H/L patients. Additionally, further research incorporating patient-derived models and single-cell transcriptomics could help elucidate the interplay between genomic alterations and tumor microenvironment factors in GC disparities. By integrating bioinformatics-driven discoveries with experimental validation, future work can refine precision medicine strategies for underrepresented populations."
We appreciate this important suggestion and have updated the discussion to emphasize the need for future experimental studies, particularly focusing on the WNT pathway.
Comment 7: The interactions between the pathways and immune genes may be an advantage.
Response: We appreciate the reviewer’s insightful comment regarding the potential interactions between oncogenic pathways and immune-related genes. While our study primarily focused on genomic alterations in TP53, WNT, PI3K, TGF-Beta, and RTK/RAS pathways in GC among H/L and NHW patients, we recognize that understanding the interplay between these pathways and the immune system could provide additional insights into ethnic-specific tumor biology and therapeutic responses.
Given that our findings indicate a higher prevalence of WNT pathway alterations in H/L GC patients (8.4% vs. 4%, p = 0.08) and a significantly lower frequency of TP53 mutations (9.6% vs. 19%, p = 0.03), it is possible that immune evasion mechanisms differ between ethnic groups. Prior studies have suggested that WNT pathway activation can suppress anti-tumor immunity by reducing T-cell infiltration and altering the tumor microenvironment. Additionally, mutations in TGF-Beta pathway components, such as SMAD4 (borderline significance, p = 0.08), have been linked to immunosuppressive signaling in GC. These findings raise important questions about how these pathways interact with immune response genes in different populations.
To acknowledge this point, we will add the following paragraph to the discussion section:
"Our findings highlight significant ethnic-specific differences in gastric cancer, particularly in TP53 and WNT pathway alterations. Given that WNT signaling is known to influence immune evasion by modulating T-cell infiltration and the tumor microenvironment, future studies should investigate the relationship between WNT activation and immune response in H/L patients. Additionally, TGF-Beta pathway alterations, including SMAD4 mutations, may play a role in shaping the immune landscape of GC. Further integration of immune profiling data, such as tumor immune infiltration scores and cytokine expression analyses, could provide a deeper understanding of how oncogenic pathways contribute to immune regulation in ethnically diverse GC patients. These insights could ultimately inform immunotherapy strategies tailored to specific populations."
We appreciate this valuable suggestion and have updated the discussion to emphasize the potential role of pathway-immune interactions in GC disparities.
|
||
|
|
||
Reviewer 2 Report
Comments and Suggestions for Authors
The manuscript “Molecular alterations in TP53, WNT, PI3K, TGF-Beta and RTK/RAS pathways in gastric cancer among ethnically heterogeneous cohorts” is primarily focused on characterizing pathway-specific mutations in TP53, WNT, PI3K, TGF-Beta and RTK/RAS signaling pathways in gastric cancer patients and determining the importance of genetic and ethnic differences while deciding the patient care. As per my recalled memory, I recently reviewed a similar article from the same group of authors entitled “WNT and TGF-Beta Pathway Alterations in Early-Onset Colorectal Cancer Among Hispanic/Latino Populations” using a similar bioinformatics analysis and appears to be an extension of similar study using a different cohort only. Although the study includes interesting findings, it relies primarily on bioinformatics analysis only using publicly available datasets, hence needs real-time validation of the data. Although the manuscript is well-written with interesting data sets, appears to be of less clinical importance. In this regard, I appreciate author’s mention that these findings need further validation using a larger, prospective studies focusing on respective populations to fully characterize GC molecular disparities, and to determine whether the observed pathway alterations translate into meaningful biological differences in tumor progression and therapy response.
Author Response
For review article
|
Response to Reviewer 2 Comments (Word file attached: Response_Reviewer_2_Comments_031325.docx)
|
||
|
1. Summary |
|
|
|
We are pleased to submit this paper and hope it will capture the interest of your readers. We have carefully addressed all your comments, highlighting the importance of this cancer disparity. Our study presents a comprehensive analysis using one of the few available genomic databases suitable for these analyses, resulting in one of the first ethnicity-focused reports on this remarkable cancer health disparity. Specifically, we examine five well-established drivers of GC— TP53, WNT, PI3K, TGF-Beta and RTK/RAS pathway alterations—within H/L populations.
Thank you very much for taking the time to review this manuscript. Please find the detailed responses below and the corresponding revisions wrote in blue font in the re-submitted files.
Reviewer 1’s responses were positive, recognizing the genetic and ethnic difference while deciding patient care to address disparities in GC outcomes within H/L populations. This feedback underscores the importance of our study and suggests that it may be a strong candidate for publication in Cancers.
Reviewer 2 writes: “The manuscript “Molecular alterations in TP53, WNT, PI3K, TGF-Beta and RTK/RAS pathways in gastric cancer among ethnically heterogeneous cohorts” is primarily focused on characterizing pathway-specific mutations in TP53, WNT, PI3K, TGF-Beta and RTK/RAS signaling pathways in gastric cancer patients and determining the importance of genetic and ethnic differences while deciding the patient care.”
We appreciate Reviewer 2’s thoughtful feedback and the recognition of our study's contribution to addressing the limited data available on this important health disparity. We believe this paper reflects our commitment to providing a comprehensive and insightful analysis of key molecular drivers in GC, focusing on the unique aspects within Hispanic/Latino populations. By leveraging the limited genomic data available today, our findings aim to contribute valuable insights into cancer health disparities and pave the way for the development of precision medicine approaches tailored to underrepresented populations.
|
||
|
Comments 1: As per my recalled memory, I recently reviewed a similar article from the same group of authors entitled “WNT and TGF-Beta Pathway Alterations in Early-Onset Colorectal Cancer Among Hispanic/Latino Populations” using a similar bioinformatics analysis and appears to be an extension of similar study using a different cohort only. Although the study includes interesting findings, it relies primarily on bioinformatics analysis only using publicly available datasets, hence needs real-time validation of the data. Although the manuscript is well-written with interesting data sets, appears to be of less clinical importance. In this regard, I appreciate author’s mention that these findings need further validation using a larger, prospective studies focusing on respective populations to fully characterize GC molecular disparities, and to determine whether the observed pathway alterations translate into meaningful biological differences in tumor progression and therapy response.
|
||
|
Response: We appreciate the reviewer’s thoughtful feedback and recognition of the significance of our findings. We acknowledge that our study is primarily a bioinformatics-driven analysis using publicly available datasets, which presents inherent limitations in terms of real-time validation. However, this study represents one of the first ethnicity-focused investigations of TP53, WNT, PI3K, TGF-Beta, and RTK/RAS pathway alterations in GC, leveraging one of the most representative publicly accessible databases that integrate both clinical and genomic data.
Despite the challenges associated with public datasets, our findings provide novel insights into molecular disparities between H/L and NHW GC patients, particularly the lower prevalence of TP53 mutations in H/L patients (9.6% vs. 19%, p = 0.03) and the higher frequency of WNT pathway alterations (8.4% vs. 4%, p = 0.08), including significantly more APC mutations (3.6% vs. 0.8%, p = 0.05). These differences suggest potential alternative oncogenic mechanisms in H/L GC patients, which could influence tumor progression and response to therapy.
As the reviewer correctly points out, further validation through larger, prospective studies is essential to fully characterize these molecular disparities and determine their clinical significance. To address this, we have updated our discussion to explicitly highlight the need for functional validation and integration of immune profiling and tumor microenvironment analyses to assess the biological impact of these pathway alterations. Additionally, given the higher frequency of WNT pathway alterations in H/L patients, we suggest that future experimental studies focus on validating the role of WNT signaling in GC tumorigenesis within this population.
To reinforce this point, we have included the following paragraphs in the discussion section:
"While this study provides novel insights into ethnic-specific genomic disparities in GC, experimental validation is needed to confirm the functional impact of the observed alterations. Given the higher frequency of WNT pathway mutations in H/L patients, future studies should focus on validating the role of WNT signaling in GC progression within this population. Functional assays, including WNT activation studies, β-catenin nuclear localization analysis, and APC gene expression profiling, could provide deeper mechanistic insights into how this pathway contributes to tumorigenesis in H/L patients. Additionally, further research incorporating patient-derived models and single-cell transcriptomics could help elucidate the interplay between genomic alterations and tumor microenvironment factors in GC disparities. By integrating bioinformatics-driven discoveries with experimental validation, future work can refine precision medicine strategies for underrepresented populations."
“One drawback of our study is the relatively small number of H/L GC patients compared to NHW patients. This limitation highlights the broader issue of insufficient genomic data availability for underrepresented populations, such as H/L, in publicly accessible databases. Nevertheless, the recent update to the cBioPortal database has expanded the available dataset, allowing our study to be among the first to leverage this increased GC cohort for analysis. Despite this challenge, our findings offer valuable in-sights into molecular disparities in GC, including a lower frequency of TP53 mutations and a higher occurrence of WNT pathway alterations in H/L patients. Furthermore, the observed borderline significant differences in PI3K and TGF-Beta pathway-related genes point to potential ethnicity-specific biological mechanisms that require further exploration. These results emphasize the urgent need for larger and more diverse genomic datasets to confirm and extend these findings, ultimately fostering more equitable cancer research and advancing precision medicine for diverse populations.”
We appreciate the reviewer’s recognition of the value of our work and have incorporated these points into the discussion to further emphasize the translational relevance of our findings. Thank you for your thoughtful feedback.
|
||
|
|
||
Reviewer 3 Report
Comments and Suggestions for Authors
This is an interesting attempt to described ethinicity dependent oncogenic pathways in gastric cancer. THe following points should be addressed for the readers:
- The study cohort looks people USA, thus definition of H/L or non-hispanic white should be clearly put in the methods. Is is self-claimed? Maybe many would be mixed actually.
- Ages, stages, and location of GC would be helpful to grasp the cohort.
- What is the referecne that shows "increasing burden of GC in H/L populations (line 123-124) " ? Ref 20-34 are on molecular analyses and those on colorectal cancer.
- The diffrence of TP53 alteration is nortworthy and are there any mutation spectrum differences detected in the database?
- Data on GC in south and middle America and ibelian populations (Portugal and Spain) are available? and any common features?
- Just cite the epidemiological inforamtion on residency or dietary lifestyle of H/L and NHW is basically similar or possibly different.
- Considerable proportion of GC have RTK amplifications (PMID 22691185) causing related signal. The authods showed mutatiaon of FGFR and others, but any amplificaion data in the database?
Author Response
For review article
|
Response to Reviewer 3 Comments (Word file attached: Response_Reviewer_3_Comments_031325.docx)
|
|||
|
1. Summary
|
|
|
|
|
We are pleased to submit this paper and hope it will capture the interest of your readers. We have carefully addressed all your comments, highlighting the importance of this cancer disparity. Our study presents a comprehensive analysis using one of the few available genomic databases suitable for these analyses, resulting in one of the first ethnicity-focused reports on this remarkable cancer health disparity. Specifically, we examine five well-established drivers of GC— TP53, WNT, PI3K, TGF-Beta and RTK/RAS pathway alterations—within H/L populations.
Thank you very much for taking the time to review this manuscript. Please find the detailed responses below and the corresponding revisions wrote in blue font in the re-submitted files.
Reviewer 3’s responses were positive, recognizing the significance of addressing cancer health disparities and highlighting the importance of our molecular focus on oncogenic pathway alterations in GC. This feedback underscores the novelty and relevance of our findings, further suggesting that the study is a strong candidate for publication in Cancers.
Reviewer 3 writes: “This is an interesting attempt to described ethinicity dependent oncogenic pathways in gastric cancer.”
We appreciate the thoughtful feedback from Reviewer 3. We believe this paper reflects our commitment to providing a comprehensive and insightful analysis of key molecular drivers in GC, with a specific focus on H/L populations. By leveraging the limited genomic data available today, our study aims to contribute valuable insights into cancer health disparities and support the development of precision medicine strategies to address these inequities. |
|||
|
|
|
||
|
Comment 1: The study cohort looks people USA, thus definition of H/L or non-hispanic white should be clearly put in the methods. Is is self-claimed? Maybe many would be mixed actually.
|
|||
|
Response: We appreciate the reviewer’s comment regarding the definition of H/L and NHW populations within our study cohort. As our analysis is based on publicly available datasets, ethnicity classifications were derived from the original database annotations. We acknowledge that self-reported ethnicity is a common approach in large-scale genomic databases, and while it provides valuable demographic insight, it does not account for potential genetic admixture.
To clarify this point, we have updated the Methods section to explicitly state that ethnicity classification (H/L vs. NHW) was based on dataset-provided annotations, which are typically derived from self-reported demographic data. While genetic admixture is a well-recognized factor in diverse populations, especially in the U.S., our study focuses on characterizing genomic differences based on the available ethnic classifications to identify potential disparities in GC at the molecular level.
Additionally, given the importance of considering genetic heterogeneity within ethnic groups, we recognize that future studies incorporating ancestry-informative markers (AIMs) or genetic ancestry analyses could provide a more precise assessment of the impact of admixture on molecular alterations in GC. To address this, we have added the following statement in the discussion section:
"Ethnicity classification in this study was based on dataset-provided annotations, which are typically derived from self-reported demographic data. While self-identified ethnicity is widely used in genomic studies, it does not fully capture the complexity of genetic admixture, particularly in H/L populations. Future research should incorporate genetic ancestry analysis to refine the understanding of molecular disparities in GC and explore how admixture influences tumor biology and therapeutic responses. Integrating ancestry-informative markers (AIMs) could provide a more precise characterization of genetic backgrounds and help tailor precision medicine approaches for diverse populations."
We appreciate the reviewer’s insightful feedback and have clarified this methodological detail in the manuscript to ensure transparency in ethnicity classification while acknowledging the need for further refinement in future studies.
Comment 2: Ages, stages, and location of GC would be helpful to grasp the cohort. Response: We appreciate the reviewer’s suggestion regarding the inclusion of additional clinical variables such as age, cancer stage, and tumor location to better characterize the study cohort. Given that our study is based on publicly available datasets, the scope of available clinical data is inherently limited. However, to enhance the clarity and comprehensiveness of our analysis, we have updated Table 1 to include more clinical data, specifically detailing the primary tumor site for both H/L and NHW patients.
This updated table provides a clearer representation of tumor distribution, showing that gastric tumors are more prevalent in NHW patients (27.1%) compared to H/L patients (10.8%), while small bowel tumors are proportionally higher in the NHW cohort (15.5% vs. 3.6% in H/L). Additionally, other tumor locations, such as bowel, jejunum, and soft tissue, were only observed in the H/L cohort. These findings highlight potential ethnic-specific differences in tumor presentation that warrant further investigation.
To address this, we have updated the manuscript by adding a more detailed description of Table 1 and incorporating a discussion of the observed differences in primary tumor site between ethnic groups. The following paragraph has been added to the table one description:
"From the cBioPortal projects mentioned above that reported ethnicity, we identified and constructed our H/L cohort, which comprised 83 samples, while the NHW cohort included 717 samples (Table 1). In terms of gender distribution, the H/L cohort consists of 45.8% males and 54.2% females, while the NHW cohort has a slightly higher proportion of males at 54.0% and 46.0% females. Regarding tumor type, all cases in both cohorts are primary tumors, with no metastatic cases reported in either group. The primary tumor site differs between the two groups. In the H/L cohort, gastric tumors are found in 10.8% of cases, whereas in the NHW cohort, gastric tumors are significantly more frequent at 27.1%. Small bowel tumors appear in 3.6% of H/L patients and 15.5% of NHW patients. Tumors in the bowel (1.2%), jejunum (1.2%), and soft tissue (3.6%) are exclusive to the H/L cohort, whereas they are absent in the NHW group. For ethnicity classification, all patients in the H/L cohort identify as Spanish/Hispanic, whereas all patients in the NHW cohort identify as Non-Spanish, Non-Hispanic. Overall, the data indicate notable differences in tumor site distribution and gender proportions between the two cohorts, emphasizing potential ethnic-specific disparities in GC."
We appreciate the reviewer’s valuable suggestion and have updated the clinical data presentation and discussion accordingly to improve the characterization of our study cohort.
Comment 3: What is the referecne that shows "increasing burden of GC in H/L populations (line 123-124) " ? Ref 20-34 are on molecular analyses and those on colorectal cancer. Response: We appreciate the reviewer’s request for a more specific reference regarding the increasing burden of GC in H/L populations. While references 20–34 primarily focus on molecular analyses and colorectal cancer, we have now incorporated a new reference (Ref #35) to directly support this statement with population-based data from a study analyzing the Texas and California Cancer Registries.
This newly added study provides relevant epidemiological evidence by showing that early-onset GC incidence rates were significantly higher among H/L individuals (1.29 per 100,000) compared to NHW (0.31 per 100,000). Importantly, this study focuses on a population similar to our catchment area in Los Angeles, CA, further reinforcing the relevance of our research in addressing GC disparities in this demographic.
To reflect this addition, we have updated the manuscript to cite Ref #35 where applicable and included this study in our reference list.
“35. Tavakkoli A, Pruitt SL, Hoang AQ, Zhu H, Hughes AE, McKey TA, Elmunzer BJ, Kwon RS, Murphy CC, Singal AG. Ethnic Disparities in Early-Onset Gastric Cancer: A Population-Based Study in Texas and California. Cancer Epidemiol Biomarkers Prev. 2022 Sep 2;31(9):1710-1719. doi: 10.1158/1055-9965.EPI-22-0210. PMID: 35732290; PMCID: PMC9444918.”
Thank you for your insightful suggestion, which has helped us strengthen the epidemiological context of our study.
Comment 4: The diffrence of TP53 alteration is nortworthy and are there any mutation spectrum differences detected in the database? Response: We appreciate the reviewer’s recognition of the noteworthy difference in TP53 alterations between H/L and NHW GC patients. Our analysis revealed that TP53 mutations were significantly less frequent in H/L patients (9.6%) compared to NHW patients (19%, p = 0.03), suggesting potential ethnic-specific molecular differences in GC development.
Regarding the mutation spectrum differences, while our study focused primarily on the overall frequency of TP53 alterations, we acknowledge the importance of analyzing specific mutation types, including missense, nonsense, or frameshift mutations, as well as hotspot variations within the TP53 gene. However, the publicly available dataset we utilized does not provide a detailed annotation of mutation spectra across ethnic groups.
To address this limitation, we have included the following statement in the discussion section:
"Our findings highlight a significant difference in TP53 mutation prevalence between H/L and NHW GC patients, suggesting possible ethnicity-specific mechanisms in tu-morigenesis. While our analysis focused on overall mutation frequency, future studies should investigate whether distinct TP53 mutation spectra exist between these popula-tions. Evaluating specific TP53 mutation types, hotspot regions, and their functional consequences could provide deeper insights into how these alterations influence tumor biology, prognosis, and treatment response. Given the current limitations of publicly available datasets in providing detailed mutation spectrum data across ethnic groups, prospective sequencing efforts focusing on diverse populations will be essential to further elucidate these differences."
We appreciate the reviewer’s insightful comment and have updated the discussion to acknowledge the importance of TP53 mutation spectrum analysis while recognizing the need for future studies to explore this aspect in more depth.
Comment 5: Data on GC in south and middle America and ibelian populations (Portugal and Spain) are available? and any common features? Response: We appreciate the reviewer’s interest in examining GC data from South and Central America, as well as Iberian populations (Portugal and Spain), and identifying potential shared molecular features. While our study specifically focuses on H/L patients from the United States, we recognize the value of broader epidemiological and molecular comparisons across Latin American and Iberian populations.
Currently, publicly available datasets for GC in South and Central America, as well as Portugal and Spain, are limited, particularly in large-scale genomic studies that integrate both clinical and molecular data, such as the dataset used in our study. As a result, this specific dataset does not include genomic data from these regions, preventing direct comparisons at this time.
We appreciate this insightful comment and agree that expanding publicly available datasets to include these populations would be valuable for future research.
Comment 6: Just cite the epidemiological inforamtion on residency or dietary lifestyle of H/L and NHW is basically similar or possibly different. Response: We appreciate the reviewer’s request to cite epidemiological information regarding residency and dietary lifestyle differences between H/L and NHW populations and their potential impact on GC risk. While our study primarily focuses on genomic differences, we acknowledge that environmental, dietary, and lifestyle factors play a significant role in GC development and may contribute to the observed disparities in molecular alterations.
Epidemiological studies have shown that H/L individuals in the U.S. have a higher prevalence of Helicobacter pylori (H. pylori) infection, a major risk factor for GC, compared to NHW individuals. Additionally, dietary patterns differ between these populations, with H/L individuals often consuming more salt-preserved and carbohydrate-rich foods, which have been associated with an increased risk of GC. Conversely, NHW individuals may have greater access to diets rich in fresh produce and lower in processed foods, potentially influencing their lower GC incidence rates.
To incorporate this perspective, we have added the following statement in the discussion section:
"Beyond genetic predispositions, environmental and dietary factors likely contribute to the observed disparities in GC between H/L and NHW populations. Prior studies have reported a higher prevalence of H. pylori infection among H/L individuals, a key risk factor for gastric carcinogenesis. Additionally, dietary differences, including higher consumption of salt-preserved foods and refined carbohydrates in H/L populations, may contribute to increased GC risk. Future research should integrate epidemiological, dietary, and genomic data to better understand the multifactorial nature of GC disparities and how these factors interact with molecular alterations in tumor development."
We appreciate this valuable suggestion and have updated the discussion accordingly to highlight the potential role of lifestyle and environmental factors in GC disparities.
Comment 7: Considerable proportion of GC have RTK amplifications (PMID 22691185) causing related signal. The authods showed mutatiaon of FGFR and others, but any amplificaion data in the database? Response: We appreciate the reviewer’s comment regarding RTK amplifications in GC, particularly the well-documented role of receptor tyrosine kinase (RTK) amplifications in driving oncogenic signaling (PMID 22691185). While our study focused on mutation-based alterations in FGFR1, FGFR2, and other RTK-related genes, we recognize that gene amplifications also play a crucial role in RTK pathway activation.
Regarding amplification data, the publicly available dataset we utilized primarily includes mutation-level genomic alterations and does not provide a comprehensive assessment of gene copy number variations (CNVs) across ethnic groups. As a result, our study does not currently include RTK amplification analysis. However, given the significance of FGFR, ERBB2, and MET amplifications in GC, we acknowledge that future studies incorporating CNV analysis would provide a more complete understanding of RTK pathway dysregulation, particularly in H/L patients.
To address this, we have included the following statement in the discussion section and the PMID 22691185 in our references section :
"While our study focused on mutation-based alterations in FGFR1, FGFR2, and other RTK-related genes, receptor tyrosine kinase amplifications are known to play a key role in GC pathogenesis. Prior studies36 have demonstrated frequent RTK amplifications, including in FGFR, ERBB2, and MET, which contribute to oncogenic signaling. However, the publicly available dataset used in our analysis does not provide comprehensive copy number variation data for these genes. Future research incorporating CNV analysis will be essential to assess the role of RTK amplifications in GC disparities, particularly in underrepresented populations such as H/L patients, and to determine their potential therapeutic implications."
We appreciate this valuable suggestion and have updated the discussion accordingly to acknowledge the importance of RTK amplifications in GC and the need for further investigation in diverse populations. |
|||
Round 2
Reviewer 2 Report
Comments and Suggestions for Authors
Thanks for taking care of reviewers' concern and modifying the manuscript accordingly.
Reviewer 3 Report
Comments and Suggestions for Authors
There is a limitation of the study, but the response is satisfactory.